# Mechanisms on How Matricellular Microenvironments Sustain Idiopathic Pulmonary Fibrosis

**DOI:** 10.3390/ijms26115393

**Published:** 2025-06-04

**Authors:** Nicole Jones, Babita Rahar, Ksenija Bernau, Jefree J. Schulte, Paul J. Campagnola, Allan R. Brasier

**Affiliations:** 1Department of Medicine, School of Medicine and Public Health (SMPH), University of Wisconsin-Madison, Madison, WI 53705, USA; nmjones7@wisc.edu (N.J.); rahar@wisc.edu (B.R.); kbernau@medicine.wisc.edu (K.B.); 2Institute for Clinical and Translational Research, University of Wisconsin-Madison, Madison, WI 53705, USA; 3Department of Pathology and Laboratory Medicine, University of Wisconsin-Madison, Madison, WI 53705, USA; jschulte2@wisc.edu; 4Department of Biomedical Engineering, University of Wisconsin-Madison, Madison, WI 53705, USA; pcampagnola@wisc.edu

**Keywords:** idiopathic pulmonary fibrosis, fibroblastic foci, extracellular matrix, hexosamine biosynthetic pathway

## Abstract

In a susceptible individual, persistent, low-level injury to the airway epithelium initiates an exaggerated wound repair response, ultimately leading to idiopathic pulmonary fibrosis (IPF). The mechanisms driving this fibroproliferative response are not fully understood. Here, we review recent spatially resolved transcriptomics and proteomics studies that provide insight into two distinct matricellular microenvironments important in this pathological fibroproliferation. First, in response to alveolar epithelial injury, alveolar differentiation intermediate (ADI) basal cells arising from Secretoglobin (*Scgb1a1*) progenitors re-populate the injured alveolus remodeling the extracellular matrix (ECM). ADI cells exhibit an interconnected cellular stress response involving the unfolded protein response (UPR), epithelial–mesenchymal transition (EMT) and senescence pathways. These pathways reprogram cellular metabolism to support fibrillogenic ECM remodeling. In turn, the remodeled ECM tonically stimulates EMT in the ADI population, perpetuating the transitional cell state. Second, fibroblastic foci (FF) are a distinct microenvironment composed of activated aberrant “basaloid” cells supporting transition of adjacent mesenchyme into hyaluronan synthase (HAS^hi^)-expressing fibroblasts and myofibroblasts. Once formed, FF are the major matrix-producing factories that invade and disrupt the alveolar airspace, forming a mature scar. In both microenvironments, the composition and characteristics of the ECM drive persistence of atypical epithelium sustaining matrix production. New approaches to monitor cellular trans-differentiation and matrix characteristics using positron emission tomography (PET)–magnetic resonance imaging (MRI) and optical imaging are described, which hold the potential to monitor the effects of therapeutic interventions to modify the ECM. Greater understanding of the bidirectional interrelationships between matrix and cellular phenotypes will identify new therapeutics and diagnostics to affect the outcomes of this lethal disease.

## 1. Introduction

### 1.1. Background

Idiopathic pulmonary fibrosis (IPF) is the most common and severe form of usual interstitial pneumonias (UIPs) [1,2]. This disease is relentlessly progressive, producing scarring, honeycombing and alveolar disruption that in advanced forms results in restrictive respiratory physiology, disrupted gas exchange and pulmonary hypertension. Consequently, patients with IPF suffer progressive dyspnea, diminished exercise capacity and reduced quality of life [3].

The incidence and prevalence of IPF has been challenging to understand due to variations in disease classifications. Using broad-based classifications, and excluding interstitial pneumonia from COVID infection, in 2005, the prevalence of IPF was ~43–63 cases/100,000 people in Westernized countries [3]. Large cohort studies in the United Kingdom have shown that the prevalence of IPF has been increasing [4], likely due to increased physician awareness, advancing age of the population and improvements in diagnosis (e.g., high-resolution CT scanning).

Although idiopathic in nature, certain patients are at higher risk for development of IPF. Of these, age is one of the dominant factors. The prevalence of IPF rises dramatically in older adults, where 1 in 1000 patients over 65 years of age carry the diagnosis. Additionally, an increase in IPF is seen in special occupations such as those serving in the armed forces who experience exposures to silicates, organopesticides, metal/wood dusts and/or combusted biofuels [5,6].

The clinical course of IPF is highly variable, a course that is affected not only by the persistence of the inciting microinjury, but also by the presence of comorbidities and the frequency of acute exacerbations (AEs). The most common comorbidities of IPF include pulmonary hypertension, gastroesophageal reflux, sleep apnea and lung cancer [3]. Comorbidities accelerate disease progression, reduce quality of life and increase mortality [7]. Separately, AEs are episodes of acute clinical deterioration in stable IPF, diagnosed by radiographic images of diffuse alveolar damage [8]. AEs occur in up to 20% of IPF patients, and are thought to be initiated by superimposed infection, aspiration and/or alveolar hemorrhage [9], and similarly are associated with rapid disease progression.

Despite its rising incidence, recent estimates show that mortality from IPF appears to be decreasing, a trend due to enhanced survival of patients diagnosed at an earlier age [10] and improved ICU management [11]. These trends notwithstanding, IPF mortality still remains unacceptably high. A systematic review of six studies reporting the mortality for IPF in 22 countries estimated that the 1-year and 5-year cumulative survival rates were 61.8% and 45.6%, respectively [12].

### 1.2. Motivation

Currently, IPF management involves supportive care, oxygen supplementation, a reduction in risk factors and use of approved anti-fibrotic therapy, such as Nintedanib and Pirfenidone [13,14,15]. However, these treatments slow, but do not reverse, the progression of the disease [16]. Lacking well-tolerated, mechanistically focused therapies, the only curative therapy for IPF is lung transplantation [3].

Thought to be merely the end result of cellular injury, increasing evidence derived from spatial proteomics and single-cell transcriptomics point to a central role that the extracellular matrix (ECM) plays in driving cell-state transitions that advance fibrosis.

### 1.3. Review Outline

This review will concentrate on the pathological cross-talk between cell-state transitioned epithelial cells and fibroblasts with an extracellular matrix. We seek to examine the characteristic features of IPF, beginning with the larger context of lung injury, introducing the concept of epithelial microinjury and reviewing the pathological changes of acute lung injury and how these overlap with those characteristics of IPF. We will then examine work that defines two distinct matricellular environments important in the pathology of IPF—the alveolar epithelial surface and fibroblastic foci. We will concentrate our analysis on the self-sustaining, autoregulatory cell–matrix interactions occurring within each. We will conclude with examining newer imaging approaches to identify cell state activation and structural changes in ECM and the potential transformative impact of these technologies in monitoring IPF progression.

## 2. Epithelial Microinjury in the Initiation of IPF

Pulmonary fibrosis develops in the setting of various injurious insults to the lung [17,18]. The classic electron micrographic studies of usual interstitial pneumonia (UIP) by Katzenstein [19] identifying the presence of granular pneumocytes in the process of re-epithelialization led to the “epithelial injury” hypothesis in IPF. This hypothesis suggests that IPF is the consequence of repetitive lower airway epithelial injuries that provoke an aberrant/exuberant wound healing response resulting in fibrosis. These exposures are termed “microinjuries” because they are not sufficient to produce clinically significant lung injury by themselves, but with repetitive exposure produce epithelial injury, senescence and stem cell failure and aberrant mesenchymal responses [20,21,22]. Factors responsible for epithelial microinjuries that have been identified include latent Epstein–Barr virus, herpesvirus infections, smoking, silicate dust and burn pit exposure [23,24]. Of importance to military personnel, burn pit emissions contain particulate matter combined with toxic gasses and heavy metals [25].

Because only a fraction of individuals exposed to these factors develop the disease, IPF is therefore a multifactorial gene by environment disease. Genetic predispositions include polymorphisms in cellular senescence pathways, telomere function, DNA repair and wound healing [26,27,28,29]. In addition, polymorphisms in mucin processing have been identified as a predisposition to fibrosis [30,31] as well as missense or deletion mutations in the surfactant protein C (SFTPC) isoforms predisposed to fibrosis. Recent attention has been paid to specific mutations in the self-folding/chaperone “BRICHOS” domain of SFPTC; these have been linked to familial childhood interstitial lung disease [32].

Other etiologies of pulmonary fibrosis include hypersensitivity pneumonitis, post-infectious fibrosis or in connective tissue diseases with pulmonary involvement; these entities will not be considered here [33,34,35,36,37].

## 3. Overlapping Features of Acute Lung Injury with IPF

The histopathological features of IPF, including fibroblastic proliferation, are seen in all forms of acute lung injuries [38,39,40]. In the simplest form of injury, one involving diffuse injury to the entire alveolar unit (Figure 1A), the alveolus heals through fibroblastic proliferation with ECM expansion in both the alveolar and interstitial compartments (Figure 1B). In this condition, fibroblast proliferation is confined to the alveolar space, representing the pathological hallmark of organizing pneumonia (Figure 1C) [38]. These forms of fibroblast proliferations are similar to fibroblastic foci seen in pulmonary fibrosis, but are self-limited and resolving [41,42].

From a histomorphologic point of view, IPF results in a well-characterized pattern of lung fibrosis with a very distinct pattern of honeycomb change [43]. In areas of honeycomb change, a fibroblastic foci (FF) located subjacent to respiratory epithelium can be identified (Figure 1D). FF are polarized zones of activated fibroblasts capped by cuboidal epithelium within an ECM rich in myxoid background. In typical FF, there are very few inflammatory cells and principally epithelial–fibroblastic structures.

Importantly, FF are typical, but not exclusive, features of IPF [44]. For example, FF can be observed in simple benign scars, or other types of idiopathic pulmonary fibrosis including patterns such as nonspecific interstitial pneumonia. As fibrosis advances and honeycomb changes progress, an acellular, collagen-rich scar is observed, especially in the peripheral lung compared to the more central portions of the lung [45].

Honeycomb change is only one feature of the prototypical pattern of fibrosis observed in IPF, known histologically as “UIP” [43]. UIP is diagnosed with a high degree of pathological certainty if five characteristic findings are present. These findings include architectural distortion with honeycomb change (Figure 1E), fibrosis in a subpleural/paraseptal location, the presence of fibroblast foci, uninvolved areas of the lung with normal architecture (patchy fibrosis) (Figure 1F) and the absence of any changes that would suggest an alternative diagnosis [43].

While UIP is a prototypical pattern of fibrosis in IPF, UIP itself can be seen as a pattern in other non-idiopathic forms of fibrosis, including connective tissue disease (rheumatoid arthritis and others) or chronic hypersensitivity pneumonitis [33,37,46]. In the setting of hypersensitivity pneumonitis, fibrosis is typically accompanied by a cellular infiltrate and granulomas. However, there is almost always evidence of prominent fibroblastic proliferation [33,34,37].

Collectively, these findings indicate that the histopathology of IPF can be viewed as a continuum of lung injury–repair processes from resolving acute lung injury to persistent matrix deposition and fibrosis.

## 4. Cell–Matrix Interactions Driving Alveolar Atypia

The earliest pathological changes so far identified in IPF can be traced to the small airways. High-resolution, three-dimensional (3D) analyses of microstructure in IPF lungs have shown a decrease in the number of terminal bronchioles and an increase in the bronchiolar wall area can be identified in regions without microscopic fibrosis [47]. In parallel, atypical cuboidal epithelial cells invade the alveoli replacing the thin squamous cell-like alveolar type 1 (AT1) population, a process known as “bronchiolization” (Figure 2, [48,49]). This repopulation of the alveoli is physiologically significant. The alveoli constitute the gas-exchanging surface of the adult human lung. In IPF, the most prominent cell populating alveolar regions are club-derived mucin-expressing basal cells [49,50]. These data, coupled with the finding that the club cell marker, secretoglobin (Scgb1a1), is upregulated in IPF [51] indicates a central role for club cell dysfunction, epithelial atypia and ECM remodeling as an early component of IPF [49].

In response to physiological senescence, AT1 cells are replenished from a reservoir of stem cells from the AT2 lineage. AT2 stem cells are maintained in a progenitor state by trophic growth factor (IL6) signals from adjacent Leucine-rich Repeat-containing G protein-coupled Receptor (Lgr)5+ or Axin2+ expressing fibroblasts [52,53]. Here, exposure of the alveolar basement membrane activates Lgr5+/Axin2+ fibroblasts to expand self-renewing AT2 cells to repopulate the alveolar surface [52,53,54,55].

Because the regenerative capacity of the AT2 stem cell population is limited, in response to diffuse injury, distinct progenitor populations within the secretoglobin (*Scgb1a1*) lineage of club cells within the distal small airway parenchyma expand to repopulate the lower airway and alveoli [56,57]. Interestingly, *Scgb1a1* precursors interact with a distinct mesenchymal population, characterized by Lgr6 expression, secreting trophic signals including Wnt and Fgf10 [58]. These activated *Scgb1a1* progenitors can be identified by their unique expression of tumor-related protein 63 (*Trp63* in mouse, or *TP63* in human) atypical keratin (*Krt)* isoforms, integrin (*Itg*) α6β4, claudin (Cldn) 4 and mechano-transductive proteins, Yes-associated protein (YAP) and transcriptional coactivator with PDZ-binding motif (TAZ). This transitional cell type is referred to as “alveolar differentiation intermediate” (ADI) cells or “damage-associated transitional”(DATP) cells [59,60]. For simplicity, here we will refer to these populations as ADI cells [56,61,62]. Importantly, these transitional ADIs can contribute ~50% to the regenerating AT2 population in bleomycin and viral injury, and therefore are a significant AT2 reservoir [56]. Importantly, persistence of this dystrophic alveolar epithelium disrupts alveolar architecture, impairing oxygen exchange [59,63,64,65]. One of the mechanisms by which this occurs is by disrupting the alveolar ECM.

Single-cell sequencing and spatial genomics studies have discovered that ADIs express signatures of three interconnected cell stress pathways—the unfolded protein response (UPR), epithelial–mesenchymal transition (EMT) and cellular senescence [59,66]. Each pathway will be discussed separately below.

### 4.1. The Unfolded Protein Response (UPR)

The endoplasmic reticulum (ER) is a site for translation-coupled three-dimensional (3D) folding of the nascent membrane, secreted and ECM proteins. Vital to the normal functioning of the cell, the ER contains membrane-bound sensors for detection of excessive protein load or accumulation aggregated/misfolded proteins. When misfolded proteins accumulate, chaperone Heat Shock Protein Family A (HSPA5/BiP) dissociates from the three major ER stress sensors activating the ER stress response. These sensors include the inositol-requiring kinase endonuclease (IRE)1, protein kinase RNA-like ER kinase (PERK) and activating transcription factor (ATF)6 (Figure 3). Each sensor is linked to activation of a transcription factor–gene expression network that coordinates the diverse arms of the homeostatic response. Collectively the UPR response degrades ER-resident proteins, degrades mRNAs, inhibits global protein synthesis and activates metabolic adaptations to promote protein glycosylation. These pathways serve to reduce the ER load; if not resolved, the cell enters programmed cell death (apoptosis).

ER stress has been mechanistically linked to IPF through genetic and animal modeling experiments. As described earlier, disrupting mutations in SFTPC processing, MUC5B and telomere function have all been linked to IPF [67]. Expression of these missense SFPTC proteins in a mouse model closely mimics the clinical course, physiological impairment, parenchymal cellular composition and biomarkers associated with IPF [68]. Mutant SFPTC expression in AT2 cells activates all three sensors of the UPR pathway (Figure 3). ER stress produces AT2 dysfunction, granulocytic inflammation and later fibrosis [68,69]. Single-cell RNA sequencing studies of epithelial populations show that ER-stressed AT2s undergo cell autonomous cellular reprogramming to produce distinct populations with homeostatic, UPR-activated and reprogrammed cell states, including ADIs [69]. The causal relationship of UPR in ADI transition was established by demonstrating that administration of an IRE1α inhibitor reduced the population of *Krt8+/Cldn4+* ADIs, as well as granulocytic inflammation. These data indicate that disruption of proteostasis is linked to cellular senescence and epithelial–mesenchymal transition (EMT).

### 4.2. Epithelial–Mesenchymal Transition (EMT)

A hallmark of the ADI transcriptional program is the expression of EMT-like gene signatures ([56,61,62], Table 1). EMT is a spectrum of partial cell-state changes of basal cells activated in response to injury [70]. Specifically, paracrine release of epithelial growth factors (TGFβ), loss of basement membrane contact, hypoxic stress and changes in ECM alone- or in combination, trigger EMT. It is important to remember that EMT is not a single-cell state, but a series of cellular de-differentiation states, characterized by various amounts of loss of apical polarity, reduced expression of cell surface cadherin, increased expression of cytoplasmic contractile proteins and activation of ECM remodeling that facilitate mucosal repair [70].

The mechanisms of TGFβ-induced EMT have been extensively studied in TP63+ human small airway epithelial cells, modeling ADI populations. Systems-level interrogations discovered that TGFβ-induced EMT is mediated by sequential cascades of autoregulatory transcription factors driving the multiple cell states in plasticity [71,72]. The transcription factors, NFκB/RelA, AP1 and SMAD3 are primary regulators that drive *TP63+* hSAECs to progress through plasticity states [71,73] inducing robust synthesis of ECM proteins, fibronectin (FN1), collagen (COL1) and matrix metalloproteinase (MMPs) [72,74]. In addition to increasing the abundance of FN1, the integrated cell stress pathways promote alternative splicing of the FN1 extra domain A (EDA-FN1), an Integrin (ITG) ligand that activates cell adhesion, fibrosis and fibroblast proliferation [75]. The sudden influx of insoluble ECM proteins activates the IRE1α-XBP1s axis of the UPR [72,74]. Once initiated, ADI cells maintain plasticity by secreting TGFβ, further promoting ER stress and metabolic reprogramming.

### 4.3. Cellular Senescence

Characterized by the permanent loss of a cell’s proliferative capability, the accumulation of senescent cells is closely coupled to progressive pulmonary fibrosis and IPF [20,76]. Senescent cells are marked by expression of senescence-associated genes, including β-gal (SA-βgal), Cyclin Dependent Kinase Inhibitor (CDKN)-1A/p21, CDKN-2A/p16 and Trp53, along with others (Table 1) [77]. This cell cycle in senescence cells is thought to limit terminal differentiation of ADI into AT1. Senescence is linked to aging, a major IPF risk factor as noted above. Mechanistically, the senescence-associated secretory phenotype (SASP) in mouse models initiates progressive lung fibrosis that closely resembles pathological remodeling seen in IPF lungs. In addition, genetic mutations in the telomere maintenance proteins activate DNA damage, producing senescence. In animal models, mutations in the telomere complex reduce proliferative response to injury, resulting in inflammation and stem cell failure [67,78]. AT2 cells isolated from mice with telomerase maintenance defects show a reduced ability to form alveospheres in vitro, demonstrating the importance of telomerase in stem cell function. Interestingly, defective epithelial telomerase expression also results in extensive secondary mesenchymal apoptosis, indicating the close bidirectional relationship between AT2 stem cells and fibroblasts within this niche [78]. Illustrating the linkage with EMT, SASP is coupled to TGFβ production, cytokine and growth factor secretion as well as production of ECM-modifying proteins [20,79]. Consequently, the UPR, EMT and senescence/SASP are coupled and coordinated intracellular signaling pathways leading to ADI persistence and ECM production.

## 5. Mechanisms of Epithelial ECM Formation

Unbiased laser-capture microdissection/proteomics studies in human IPF have found the presence of unique ECM composition in bronchiolized, non-fibrotic alveoli [80]. Here, 45 unique differentially expressed ECM proteins were identified compared to control alveoli [80]. These proteins included Serpin Family B Member 12 (SERPINB12), and fibrillin-1 (FBN1), proteins associated with cystic fibrosis and connective tissue disease. These findings further support the concept that active ECM remodeling occurs in alveoli with atypical re-epithelialization.

In TP63+ ADI cells, the mechanisms of TGFβ-inducible ECM remodeling have been studied at the systems level. Here, TGFβ stimulation induces coordinate cell state plasticity through upregulation of the IRE1α-XBP1s arm of the ER stress response [81] and the IKK-NFκB pathway [82], leading to activation mesenchymal transcription factors Snail Family Transcriptional Repressor (SNAI)-1 and Zinc Finger E-Box Binding Homeobox (ZEB). This cascade leads to the production and secretion of ~100 N-glycosylated ECM proteins [81]. The composition of the subepithelial ECM has been defined by matrisome analysis of cultured TP63+ ADI cells [81], LC microdissection studies of human lung [80,83] and quantitative detergent solubility profiling of mouse lung [84]. These studies have shown that although the epithelium is not normally a major producer of ECM, in response to injury, infection or stimulation by TGFβ-like growth factors, ADI cells actively form and remodel ECM through the coupled UPR, EMT and senescence pathways discussed above. This mechanism of how epithelial cells secrete glycosylated ECM components was demonstrated by discovery of metabolic reprogramming through the inducible hexosamine biosynthetic pathway (HBP).

The HBP is an intracellular nutrient-sensing pathway that metabolizes glucose, glutamine, acetyl-CoA and UTP into the nucleoside sugar, uridine diphosphate-N-acetyl glucosamine (UDP-GlcNAc; Figure 4). Glutamine:fructose-6-phosphate amidotransferase 2 (GFPT2) catalyzes the first committed- and rate-limiting step in the HBP pathway converting fructose-6-phosphate and glutamine into glucosamine-6-phosphate [85]. UDP-GlcNAC is the obligate substrate for the OGT-mediated Glc-NAcylation and subsequent protein O- and N-glycosylation as well as production of the proteoglycan hyaluronan [86]. Protein N-glycosylation is important in nascent protein folding and ER quality control, ER-associated apoptosis and secretion of ECM proteins. N-glycosylation enhances the stability and structural integrity of major alveolar protein constituents, including FN1, COL and laminin (LAM). Importantly, glycosyltransferases and glycosidases, enzymes responsible for glycosylation, show dysregulated activity in IPF, leading to altered protein–glycan structures [87]. For example, the alpha-(1,6)-fucosyltransferase (FUT8) transfers fucose to the innermost GlcNAc residue on N-glycans, and plays a crucial role in the development of IPF by regulating TGFβ-induced alveolar senescence [88]. More work understanding the biochemistry of protein N- and O-linked glycosylation will be important for fully understanding the biochemical pathways underpinning matrix remodeling in IPF.

## 6. Changes in ECM Composition and Stiffness Promote ADI Persistence

The AT1 cell interacts with a basal lamina and thin ECM, which provides structure and elasticity to this gas-exchanging unit. The alveolar ECM is an interlocking mesh of collagens, elastic fibers and glycoproteins and matrisome-associated proteins that play a critical role in the maintenance of epithelial polarization and signaling. The basal lamina is a self-assembling ECM rich in LAM, proteoglycans and COL VII that binds to cell-surface integrins to anchor its cytoskeleton to the basal lamina. With activation of the integrated cell-stress pathways causing EMT/UPR/senescence, FN1, COLs and ECM-modulating MMPs are secreted into the basal lamina, causing fibrillogenic change (Figure 5). This fibrillogenic ECM inhibits differentiation, promotes increase in ECM stiffness and stimulates EMT through multiple self-reinforcing pathways [79]. In particular, FN1 is one of the most highly upregulated proteins produced in response to TGFβ and plays an integral part in nucleating ECM, sequestering growth factors and enhancing matricellular stiffness [89,90]. Upregulated in ADI, the ITG isoforms αvβv/αvβ6 bind to regions of FN1 changing conformation of the latency-associated protein to activate local TGFβ release [91]. ECM proteoglycans are fragmented by TGFβ-induced MMPs; these fragments bind and stimulate CD44, stimulating EMT. Enhanced contractility releases lysyl oxidase (LOX), responsible for oxidative deamination of peptidyl lysine residues in COL and elastin to further COL crosslinking, activating mechano-transduction in ADIs through YAP/TAZ signaling. Both N- and O-glycated forms of FN1 are produced by EMT-activated ADI cells [92].

FN1 secretion plays a central role in promoting alveolar ECM remodeling. FN1 provides a scaffold for COL deposition, which is stabilized by cross-linking enzymes like transglutaminases and LOX, enhancing tissue stiffness [93]. FN1 requires extensive glycosylation to function effectively in cell adhesion, migration, growth and differentiation [94]. Its interactions within the ECM and with integrin proteins depend on proper glycosylation [95]. Aberrant glycosylation of FN1 alters ECM integrity and signaling pathways [96]. Using a specific N-glycosylation-specific LC-MS/MS assay, we discovered that TGFβ induces a ~900-fold upregulation of N-glycosylation of FN1 on Asn430 and Asn528 residues. LAM, a key structural protein in alveolar basement membranes [97], undergoes glycosylation for proper folding, stability and receptor interactions. LAM glycosylation is required for binding and signaling to its cognate ITG and dystroglycan receptors. Dysregulated LAM expression has been linked to altered TGFβ signaling and progression of fibrosis in the bleomycin mouse model [98]. Finally, the stability and function of COL depend on post-translational modifications, including hydroxylation and glycosylation. Defective collagen glycosylation has been linked to impaired ECM integrity in fibrosis [96].

Not only may these changes affect mechanical stability of the alveolar structure, but also modulate alveolar cell function. Exposure of primary alveolar cells to FN1 induces EMT via ITG αvβ6 [99]. In a follow-up study, deletion of ITG α3 in lung epithelium prevented mice from developing fibrosis and led to a decrease in myofibroblast population and COL I expression after bleomycin injury [100]. Increased mechanotransduction from a stiff ECM reduces epithelial proliferation and repair capacity, promoting a vicious cycle of fibrosis and injury [101].

A discussed below, real-time imaging of EDA-FN1 deposition holds promise in determining disease activity. These findings suggest an intricate relationship between senescence, ECM deposition and cell state transition fundamental in promoting or sustaining disease. These findings strongly suggest that alveolar EMT is regulated by growth factors and ECM interactions through ITG signaling.

## 7. Matricellular Organization of Fibroblastic Foci (FF)

FF are polarized structures with regional collections of macrophages, aberrant basaloid epithelial cells and distinct types of fibroblasts (Figure 1, [102]). Here the leading edge actively produces COL- and FN1-rich ECM. As noted earlier on the formation of multiple cysts, “honeycombing” subpleural cystic structures is an end-stage manifestation of parenchymal destruction and loss of architecture. Also noted earlier, FF are a ubiquitous feature of lung injury; however, their abundance and persistence are characteristic of IPF and can be correlated with disease progression and mortality [103]. Basal lamina remnants on the interstitial side of FF suggest that their origins lie in previously normal airspaces [80,104].

### 7.1. Aberrant Basaloid Epithelial Cells

Responsible for the majority of ECM deposition in fibrosing lung diseases, FF have been intensively investigated in the pathogenesis of IPF. Importantly, FF are associated with a cuboidal epithelial “cap” (Figure 1). These cuboidal epithelial cells have been analyzed by single-cell and spatial proteomics studies and noted to be composed of aberrant phenotype, “basaloid”, epithelial cells with similar features to ADIs [56,102,105]. These aberrant basaloid cells express both AT1 (Aqp5) and low levels of AT2 (SFPTC) markers [80], as well as a unique pattern of basal epithelial markers including TP63, KRT17, LAM-B3 and -C2, but not other established basal cells markers as KRT-5 or -15. In a manner reminiscent of ADI, aberrant basaloid cells exhibit profiles of the integrated stress response activation, expressing EMT signatures of mesenchymal contractile proteins (VIM), ECM (FN1, COL1A1, TNC) and cellular senescence markers (CDKN1A, CDKN2A, CCND1).

The function of aberrant basaloid cells in driving FF behavior is not fully understood, but currently thought to involve the production of trophic signals for maintaining myofibroblast activity and ECM modifications. Evidence suggests these aberrant basaloid cells are engaged with TGF signaling with distinct mesenchymal populations within FF [102,105]. Specifically, aberrant basaloid cells express TGFβ2, with the underlying myofibroblasts expressing TGFβ1 and TGFβ3, suggesting the presence of bidirectional mesenchymal–epithelial trophic signaling pathways [80]. Cell signaling analysis indicates that TGFβ1, IL1β and ApoE signaling were predominant epithelial–mesenchymal signals in FF [105]. In this latter signaling analysis, the presence of COL-ITG signaling from the ECM signals was also identified.

### 7.2. Fibroblast and Myofibroblast Populations in FF

Fibroblasts within FF are complex and heterogeneous, arising from multiple stromal populations from resident fibroblasts and circulating fibrocytes from the bone marrow [60,106,107]. Unique to IPF is a fibroblast population characterized by high levels of hyaluronan synthase (HAS)-1 and -2 [60]. These HAS1^hi^ cells carry signatures of IL4/IL13 signaling and EMT markers. In situ, HAS1^hi^ fibroblasts colocalize with aberrant basaloid epithelial cells and subpleural regions rich in COL1A1, consistent with an invasive phenotype [105]. Relating to the potential paracrine signaling by aberrant basaloid cells, we note that HAS expression in fibroblasts is highly induced by IL1β, perhaps explaining the close spatial relationship between aberrant basaloid cells and HAS1^hi^ fibroblasts [108]. Hyaluronan metabolism is of particular interest in IPF. HA is high molecular weight glycosaminoglycan that functions in physicochemical properties of ECM as well as signaling via CD44 receptors, which is important in activating cell adhesion and migration. HA fragmentation is implicated in CD44-dependent fibroblast invasive properties and IPF [109].

A distinct population of mesenchyme in FF are synthetic myofibroblasts, cells that express COL1 and smooth muscle Actin alpha2 (ACTA2). Myofibroblasts have distinct regional distribution from that of HAS1^hi^ fibroblasts, where they are enriched in the FF core [105] and in subepithelial regions around large airways [102]. Mechanistically, myofibroblasts arise from a coordinated epigenetic transition of resident fibroblasts and circulating fibrocytes initiated by TGFβ and ECM signaling similar to that of EMT. These stimuli regulate microRNA abundance and activate cytoplasmic NADPH oxidase (Nox4) to effect synthetic myofibroblast transition; this pathway has been extensively reviewed previously [110]. Transcriptomic studies highlight the central role of TGFβ signaling in myofibroblast transition. Other key mediators such as TP53, SMAD3, BMP7, MRTFB, TEAD, GLIS1 and APOE regulate senescence, Notch and Wnt pathways; apoptosis; and cell migration [105,111]. Myofibroblast transition produces high levels of ECM production, driving persistent fibrosis and tissue scarring [112]. IPF myofibroblasts exhibit altered gene expression profiles, enhanced proliferative capacity [111,113], resistance to apoptosis, anchorage-independent growth and defective translational control compared to normal lung fibroblasts [114,115].

### 7.3. The Matrix of FF

The leading edge of FF produce highest synthesis rates of COL and pathogenic airspace remodeling (Figure 6, [116]). Laser capture microdissection with LC-MS proteomics has characterized the quantity and composition of matrix in IPF FF. Reflecting its distinct matricellular composition, the matrix of FF is distinct from both alveolar ECM and a mature scar in IPF [80]. Laser capture/microdissection studies have identified over 100 proteins unique to the IPF ECM. These include COL isoforms important in fibrillogenesis, lysyl hydrolases responsible for ECM crosslinking (PLOD/P4HA) and MMPs catalyzing ECM remodeling (MMPs-14 and 2; Ref. [80]). Proteins unique to FF vs. pre-fibrotic alveoli and a mature scar were identified; these include FN1, tenascin C (TNC), SERPINH1 and versican (VCAN). A systematic time course study of the matrisome in the bleomycin mouse model also provided evidence for substantial ECM remodeling, where 435 matrisome and 264 matrisome-associated proteins were identified. Bleomycin treatment induced shifts in detergent solubility of the matrisome-associated proteins indicating that substantial remodeling and signaling was occurring [84]. Lacking spatial resolution, and acknowledging the limitations of the mouse model, the relevance of these findings to human FF are presently unclear.

MMPs are a family of Zn-containing endopeptidases with specificity to COL that play key roles in ECM remodeling. A well-established marker of IPF progression [117], MMP-7, contributes to ECM degradation by activating other MMPs, regulating TGFβ and influencing osteopontin, thereby promoting fibrosis [118]. MMP-9 degrades COL IV and remodels the basement membrane but also increases inflammation and endothelial permeability. Tissue inhibitors of metalloproteinases (TIMPs) counteract MMP activity, with TIMP-2 predominating in FF, stabilizing the matrix and promoting ECM accumulation [119].

With FF persistence, the matrix in FF undergoes a dramatic switch where distinct patterns of glycoproteins, proteoglycans, COL and ECM regulators are enriched. Areas of acellular mature scar are formed in IPF (Figure 1 and Figure 6). In IPF lungs, COL III is the predominant component in areas of alveolar septal fibrosis, while COL I is more abundant in regions of mature fibrosis [120]. Upregulated LOX expression is observed in IPF [121]. LOX is a copper-dependent extracellular enzyme that catalyzes the oxidative deamination of lysine residues in collagen, producing aldehyde groups essential for covalent cross-link formation [122]. These cross-links increase ECM stiffness, contributing to fibrosis and tumor progression. A mature scar shows a substantial increase in stiffness; atomic force microscopy measurements show that FF are relatively soft with a Young’s modulus of 1.5 kPa, whereas a mature scar is stiffer with a modulus of 9.0 kPa [123].

### 7.4. ECM Stiffness and Myofibroblast Mechanotransduction

ECM stiffness significantly influences cell behavior, impacting fibrosis progression [124]. Normal lung tissue has a shear modulus of ~0.5 kPa [101]. Increased stiffness enhances fibroblast traction forces, activating TGFβ and driving sustained activation and matrix deposition. ECM stiffness also affects cellular viability/transition. Epithelial cells respond to stiffness-dependent TGF-β signaling by undergoing apoptosis on soft substrates and EMT on stiffer matrices [125].

Mechanotransduction pathways, particularly the YAP/TAZ pathway, activated in ADI and myofibroblasts, mediate cellular responses to ECM stiffness. On soft substrates, the Hippo pathway retains YAP/TAZ in the cytoplasm through MST1/2 and LATS1/2 kinases. Cells sense stiffness via ITG-containing focal adhesions, linking ECM to the cytoskeleton [126]. By contrast, high ECM stiffness suppresses Hippo activity, facilitating YAP/TAZ activation and nuclear translocation. Additional signaling pathways, including FAK and Rho GTPases, further modulate mechano-transduction [127]. Consequently, nuclear YAP/TAZ interact with EA domain transcription factor 1 (TEAD) transcription factors to drive proliferation, survival and ECM production. These processes create a self-sustaining fibrotic cycle, reinforcing persistent myofibroblast activation and reinforcing ECM stiffness in IPF.

## 8. Imaging of ECM to Advance Novel Anti-Fibrotics in IPF

Radiographic assessments (via high-resolution computed tomography (HRCT)) and respiratory impairment (via pulmonary function tests (PFTs)) are the most commonly employed methods for clinical evaluation of disease progression [128,129]. These metrics offer insights into overall changes between exams conducted months apart, in an effort to characterize the shifts in radiographic opacities through high-resolution CT and assess declines in forced vital capacity via PFTs. As critical as these methods are in the current clinical management of IPF, they are not able to distinguish stable from progressive IPF, nor are they able to characterize between the heterogenous molecular mechanisms, which, as new therapeutic options become available, may lead to differential treatment approaches.

Molecular imaging has emerged as an advanced imaging method to non-invasively detect biological processes through the use of molecular probes. Positron emission tomography (PET) and single-photon emission computed tomography (SPECT) imaging employ radiolabeled probes to sensitively detect a molecule, pathway or receptor of interest down to the picomolar to nanomolar range. Due to their low anatomical resolution, these functional imaging methods are often combined with high-resolution imaging modalities, most notably CT, to provide detailed anatomical information that can be overlayed during analysis. While other molecular imaging techniques exist, including gadolinium-based magnetic resonance imaging (MRI) tracers, the PET/CT integrated approach is particularly well-suited for real-time assessment of IPF disease activity given the heavy reliance on CT modality for high-resolution detection of radiographic changes in the clinical setting.

Development of molecular imaging probes requires identification and validation of targets that are specific and highly upregulated in fibrosis, ideally specific to fibrogenesis. Consequently, lack of molecular imaging probe accumulation in stable, non-progressive disease provides an added advantage during characterization of disease activity. Beyond tracer uptake within the regions of interest, background tissue uptake is also an important consideration. High accumulation in background tissues could interfere with signal of interest, although this could be mitigated during analysis depending on the spatial localization of the background tissue. Here, we provide a brief summary of several notable molecular imaging targets and probes currently under investigation.

### 8.1. Targeting Dysregulated Epithelia

ITG αvβ6, an epithelial-restricted receptor that activates TGFβ, is highly expressed in fibrosis and shown to be significantly associated with decreased survival in patients with IPF [130,131]. Several categories of αvβ6 detecting peptide-based PET probes are currently under clinical investigation [132]. As part of the PETAL study, the authors showed that the lung uptake of the linear tracer, [^18^F]-FB-A20FMDV2, was significantly increased in subjects with IPF and connective tissue disease-associated pulmonary fibrosis compared to healthy controls [133]. In addition, this PET tracer was utilized for confirmation of target engagement in patients treated with an inhaled inhibitor of αvβ6 [134]. [^18^F]FP-R01-MG-F2, the cystine knot αvβ6-targeting peptide (knottin) PET probe, also accumulated in the lungs of IPF subjects, as well as mice subjected to bleomycin-induced pulmonary fibrosis, compared to controls [135]. Interestingly, the authors of these studies found differential uptake of these αvβ6-targeting PET tracers in the GI-tract, with higher uptake seen with linear peptides [132].

### 8.2. Targeting Activated Fibroblasts

Molecular imaging that targets activated fibroblasts and myofibroblasts offers another approach for detection of active disease. Fibroblast activation protein (FAP), a type II transmembrane serine protease, is specifically and differentially upregulated on stromal cells within FF, while not being expressed in the normal lung [80,136]. Radiolabeled small molecule inhibitor of FAP, [^68^Ga]Ga-FAPI-46 accumulated in the lungs of bleomycin-treated mice at both 7- and 14-days post-bleomycin treatment. On the other hand, CT imaging only detected a change in radiodensity during the 14-day time point, suggesting that this PET tracer was able to detect fibrotic disease activity ahead of apparent architectural changes found on CT imaging [137]. In another study, both [^68^Ga]Ga-FAPI and [^18^F]-FDG detected bleomycin-induced fibrosis, and their uptake decreased upon treatment with pirfenidone. However, the peak uptake differed between the tracers with accumulation of [^18^F]-FDG and [^68^Ga]Ga-FAPI being the highest on day 14 and 21 post-bleomycin, respectively, suggesting that [^68^Ga]Ga-FAPI does indeed target a different mechanism than [^18^F]-FDG [138]. Furthermore, the total and mean standardized uptake value (SUV) of [^68^Ga]Ga-FAPI-04 was increased in the lungs of IPF subjects, as well as those with non-IPF ILD, in comparison to healthy controls. SUVtotal correlates with clinically relevant changes in pulmonary function testing, including percent-predicted baseline forced viral capacity (FVC) and diffusion capacity (DL_CO_ [139]).

### 8.3. Detecting Active ECM Deposition

While detection of aberrant cellular phenotypes provides important molecular insights into disease pathogenesis, direct assessment of active ECM deposition offers another viewpoint into lung fibrosis disease progression. EP-3533 is a gadolinium-labeled peptide with low micromolar affinity for COL I that demonstrated the capacity for detecting lung fibrogenesis in mice as early as 5 days post-bleomycin via MRI [140]. This peptide was subsequently adapted for PET imaging using the [^68^Ga]Ga-CBP8 probe. Accumulation of both EP-3533 and [^68^Ga]Ga-CBP8 tracers correlated with lung COL content, suggesting that this probe may provide accurate assessment of COL deposition [140,141]. This probe was subsequently approved for clinical studies and has so far demonstrated increased uptake in fibrotic regions, as well as those that appear normal on CT, indicating potential for predicting early disease activity [142]. Beyond COL I itself, there are other notable markers that participate in collagen assembly and damage that may be targets for molecular imaging to detect early disease activity, including FN1, oxidized COL and allysine [143,144,145,146]. In particular, a FN1-targeted peptide, PEG-FUD, was recently found to preferentially bind to regions of nascent fibrosis, including FF, in the IPF lung ex vivo, as well as target early phases of the bleomycin-induced murine lung injury and fibrosis [143,147]. Given the critical role of FN1 in COL assembly, this approach may be another promising strategy for early detection of fibrogenesis using molecular imaging approaches.

## 9. Optical Imaging Approaches

The development of an imaging approach that can provide additive diagnostic/prognostic information from lung biopsies or could be used non-invasively to provide longitudinal information would be a clear advance for the field. Optical imaging approaches provide sensitive and specific readouts with sufficient resolution to identify changes across spatial scales in the ECM structure.

A major roadblock in obtaining this information is the lack of adequate techniques to probe different levels of collagen structure (e.g., fibrillar and supramolecular) with high sensitivity and specificity, specifically in a 3D context including other ECM components (elastin and FN1). This is especially true in examining regions near FF. Such regions are thought to be at the leading edge of ECM remodeling but the dynamics of their formation in relationship to the overall fibrotic process are unknown [148,149,150]. Even in conjunction with histology, established clinical imaging techniques such as HRCT cannot provide quantitative, objective assessment at this required level of structural analysis.

To solve this problem, we have used Second Harmonic Generation (SHG) imaging microscopy to selectively image the remodeling of the collagen in IPF (and other diseases) [151,152,153,154,155,156,157]. This modality provides intrinsic 3D imaging with high resolution (~0.5 microns) and affords imaging depths into tissues up to a few hundred microns [158,159]. SHG does not utilize exogenous stains and directly visualizes the COL assembly and is sensitive to changes therein in diseased states [151,152,160,161,162,163,164,165,166,167,168,169]. Additionally, through polarization resolved analyses, macro/supramolecular details on the collagen assembly can be elucidated, where these include the alignment of molecules within fibrils as well as the composition of collagen isoforms (e.g., COL I and III) within the fibrils as well as the helical structure [155,170]. Most importantly, this level of structural information cannot be obtained through H&E histology, immunostaining or by autofluorescence, which are primarily sensitive to the concentration and not the structure. While we have used these polarization techniques to characterize IPF tissues, here we will comment on strategies probing fibrillar morphology.

### 9.1. Classification of Normal and IPF Fibrillar Morphology

We have successfully used machine learning for classification of normal and IPF lungs using 2D wavelet transforms based on the respective morphology revealed by SHG microscopy. Using principal component analysis (PCA) of 2D wavelet transforms, we found accuracies of >95% [153]. We then adapted the procedure to 3D analysis. This level of data is most often not exploited in SHG microscopy, despite the intrinsic optical sectioning. Using ROC analysis of true positives vs. false positives, accuracies based on an area under the curve of 85% were obtained in differentiating normal and IPF lungs.

### 9.2. Multiphoton Microscopy Imaging Multiple of ECM Components

We specifically probed both collagen (SHG) and elastin (two-photon excited (TPE) autofluorescence), where these contrasts were simultaneously excited at the same wavelength (890 nm) and spectrally isolated in separate channels. As the elastin contrast is linearly proportional to the concentration, and SHG is a merged effect of the square of the collagen concentration and its organization, it is not possible to determine their actual molecular ratios. However, we analyzed the volumetric ratio of elastin and collagen using the well-documented method, [E_V_ − C_V_]/[E_V_ + C_V_], where E_V_ and C_V_ represent elastin and collagen voxel volumes, respectively [171,172]. The representative background-corrected two color images of the SHG (green) and elastin (blue) TPEF for normal and IPF tissues are shown in Figure 7a,b, respectively, where the organization of the collagen and the elastin are both dramatically different in these cases. Specifically, the elastin in the normal tissues is organized within the confines of the collagen, whereas in the diseased tissue the elastin is disorganized and not exclusively intermingled with the collagen fibers. The elastin/collagen index derived from all parenchymal imaging stacks in normal and IPF tissues is shown in Figure 7. The normal tissues were more elastic relative to collagen than the diseased tissues indicative of an altered matrix composition. This finding determined by optical microscopy is consistent with the mechanical consequences of the disease.

### 9.3. SHG Imaging of Spheroids

A barrier to investigating ECM changes in IPF has been the absence of translational preclinical human relevant models. Jones and co-workers established a long-term 3D model system (up to 60 days) where fibroblasts derived from human lung control or IPF tissue produce structured ECM including fibrillar collagens and proteoglycans. In culturing IPF fibroblasts, there are clear histochemical similarities in organization to fibroblastic foci, the site of active fibrillogenesis in IPF, and proteomic characterization further confirms ECM comparability to ex vivo tissues [173]. The model enables the evaluation of the collagen structural analysis. Figure 8 shows examples of IPF spheroids (42 days) where crosslinking was inhibited and promoted by treatment with PXS-5-5120 and IOX2 (increases LOX), respectively. The former resulted in a decreased fiber size and SHG intensity, whereas promotion enhances brightness [121].

### 9.4. Optical Coherence Tomography (OCT) Imaging of IPF

SHG is a microscopic technique, where the individual fields of view (FOVs) needed to resolve collagen fibers and satisfy the Nyquist criteria are typically limited to a ~200 microns. This size is far smaller than the typical honeycomb structure observed in IPF by HRCT (several mms) and is just below the typical size of microscopic honeycombing (several hundred μm). To address this problem, SHG imaging has been combined with optical coherence tomography (OCT) on the same microscope platform. OCT is typically performed at low mag/NA and is ideal for rapid probing of large areas. This is important in IPF, as there is significant heterogeneity in the tissue. There have been recent reports of using OCT to image lungs, where we specifically note that Suter et al. showed that in vivo OCT could visualize the IPF honeycombed structure [174,175], where these structures were around a few hundred µms to a few mms. Using simple visualization without quantification, this work recapitulated image data seen in low power histology and provided evidence that OCT can be used for this purpose. More recently, PS-OCT, which is sensitive to alignment, was successfully used in conjunction with machine learning to diagnose IPF tissues in vivo [176].

## 10. Practical Implications

Recognizing the important autoregulatory component of cell–matrix interactions has significant implications in therapeutic approaches and treatment monitoring. Here, we identify the integrated epithelial cell stress response that drives exuberant wound repair response in predisoposed individuals. Therapeutics that modify key regulatory proteins driving the UPR, EMT and HBP may be viable for drug development. Moreover, the ability to detect the presence of activated epithelium, myofibroblasts and pathological ECM remodeling provides new opportunities for disease monitoring.

## 11. Summary and Conclusions

In summary, we review two major matricellular environments that play important roles in the progression of IPF. Transitional cells with activated UPR, EMT and senescence pathways repopulate the alveoli, producing bronchiolization and alveolar dysfunction. This process is associated with ECM remodeling and enhanced stiffness, which play an important role in the maintenance of cells in a de-differentiated state. FF are an invasive multicellular entity composed of a matrix associated with atypical basaloid epithelial cells and several unique populations of fibroblasts. FF are responsible for the majority of alveolar destruction and a mature scar in IPF. We highlight advancements in MR, PET and optical coherence imaging that will enable mechanistic and translational studies to advance therapeutic approaches to disrupt the vicious ECM-cellular signaling.

## Figures and Tables

**Figure 1 ijms-26-05393-f001:**
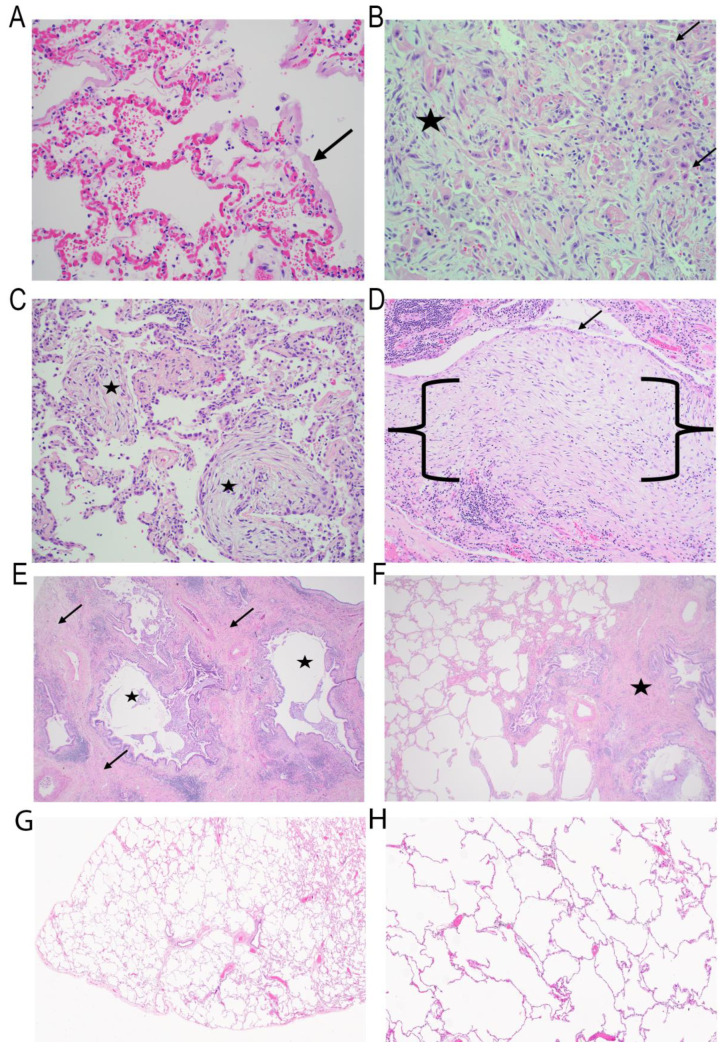
ECM microenvironments in IPF. Histological images of human pathological samples acquired during routine care. (**A**) Diffuse alveolar damage, exudative phase. Alveolated lung parenchyma showing capillary congestion and thick pink hyaline membranes (black arrow) linking alveolar spaces consistent with diffuse alveolar damage, the prototypical response seen in the lung to acute lung injury (H&E, 200×). (**B**) Diffuse alveolar damage, organizing phase. Lung parenchyma showing haphazard proliferation of fibroblasts (black star) filling both alveolar and interstitial spaces. Large individual cells (black arrows) represent hyperplastic type 2 pneumocytes (H&E, 200×). (**C**) Organizing pneumonia. Intra-alveolar fibroblastic proliferations (Masson bodies, black stars) are the pathologic hallmark of organizing pneumonia. Organizing pneumonia is a nonspecific response to numerous types of lung injury (H&E, 200×). (**D**) Fibroblast focus. The fibroblast focus (within black brackets) shows parallel arrangement of fibroblasts, typically oriented in the same direction as the overlying epithelium (black arrow). The fibroblast focus typically lacks an inflammatory infiltrate and a collection of lymphocytes can be seen excluded from the fibroblastic proliferation at the bottom of the image (H&E, 100×). (**E**) Honeycomb change. End-stage lung showing cystic spaces lined by respiratory epithelium (black stars). Collagen rich fibrosis (black arrows) intervenes between the cysts. These are the classic features of honeycomb change (H&E, 20×). (**F**) Usual interstitial pneumonia. Usual interstitial pneumonia typically shows a patchy distribution of fibrosis with involved and uninvolved areas. The right side of the image shows an area of advanced fibrosis with honeycomb change (black star) and the left an area of relatively normal lung with preserved alveolar architecture (H&E, 20×) (**G**) Normal lung for comparison. The lung shows no evidence of fibrosis with normal distribution of septal structures and bronchovascular pairs (H&E, 10×). (**H**) Normal lung for comparison. Thin delicate alveolar septations without evidence of fibrosis (H&E, 100×).

**Figure 2 ijms-26-05393-f002:**
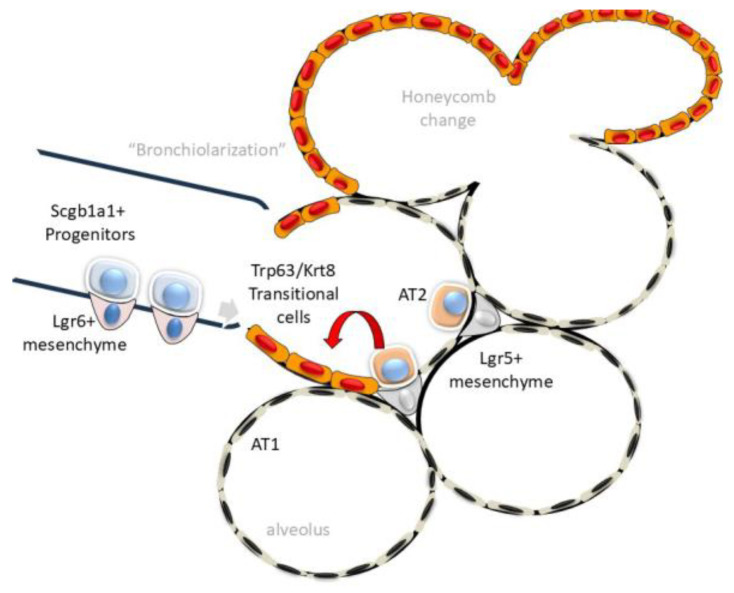
Alveolar atypia from ADI in IPF. Schematic of the transition of the epithelial injury–repair response in the bronchiole–alveolar junction. Under resting conditions, small airway progenitors derived from *Scgb1a1+* epithelium are maintained in a progenitor state within a niche with *Lgr6+* mesenchyme. In response to lower airway injury, growth factors and matrix metalloproteinases are released to initiate *Trp63+* transitional cell-repair. These are migratory ADI cells and have activated UPR, EMT and senescence pathways, whose persistence leads to failed redifferentiation. In parallel, AT2 cells interacting with the *Lrg5+* mesenchyme also contribute to atypical ADI cells. A region of cystic degeneration “honeycombing” is illustrated at the top of the figure.

**Figure 3 ijms-26-05393-f003:**
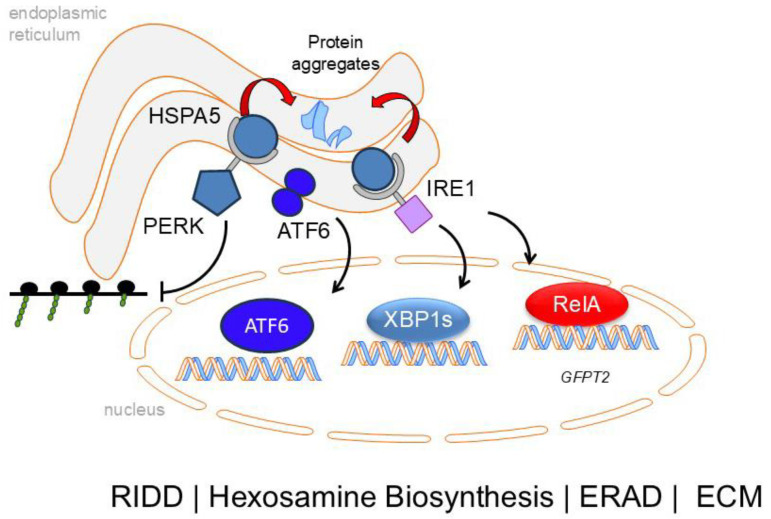
The unfolded protein response. Protein processing in the epithelial cell is monitored by ER sensors. Aggregation of ECM proteins within the ER is bound by chaperonin HSP5a/BiP, preventing their aggregation and assisting in proper folding. Dissociation of HSP5a from IRE1α enables IRE1α to oligomerize and autophosphorylate. The RNase domain cleaves and activates XBP1 mRNA, leading to the production of the spliced transcription factor XBP1s, in which IRE1α kinase induces JNK and NFκB signaling. XPB1s activates hexosamine biosynthesis to support secretion of ECM remodeling proteins, restoring proteostasis. Dissociation of HSP5a from PERK activates a distinct pathway, resulting in phosphorylation eukaryotic initiation factor 2 alpha (eIF2α) at its serine 51 site. The result of this activation is the inhibition of global translation to reduce the burden of newly synthesized proteins entering the ER. ATF6 activation, on the other hand, requires its transport to the Golgi, where it is sequentially cleaved. Abbreviations: RIDD, regulated, IRE1α-dependent (RNA) decay; ERAD, ER-associated degradation.

**Figure 4 ijms-26-05393-f004:**
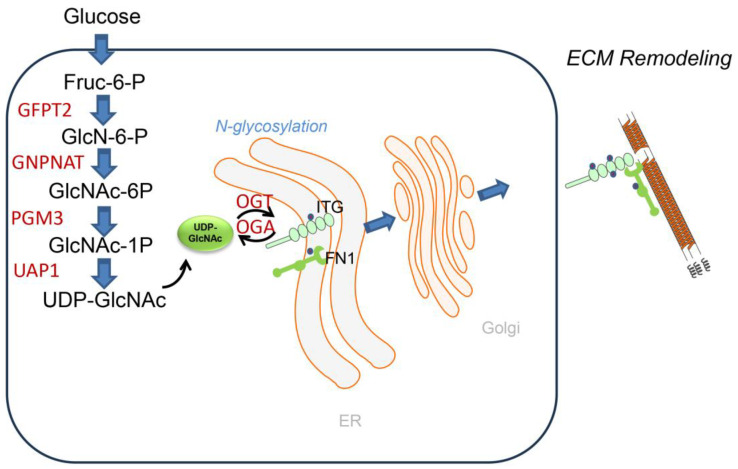
HBP pathway in ECM glycosylation. Schematic diagram of the intracellular shunting of glucose into the UDP-GlcNAc molecule. Key enzymes of the HBP, glutamine-fructose-6-phosphate transaminase (GFPT)-1 and -2, glucosamine-phosphate N-acetyltransferase (GNPNAT) and phosphoglucomutase (PGM3) are upregulated in the TGFβ-induced EMT. GFPT converts d-fructose-6-phosphate (Fru-6-P) and l-glutamine to d-glucosamine-6-phosphate (GlcN-6-P) and l-glutamate. GlcN-6-P is an essential precursor of uridine 5-diphosphate-N-acetyl-d-glucosamine (UDP-GlcNAc), a rate-limiting substrate of the O-GlcNAc transferase (OGT) in the HBP, a pathway required for glycoprotein formation.

**Figure 5 ijms-26-05393-f005:**
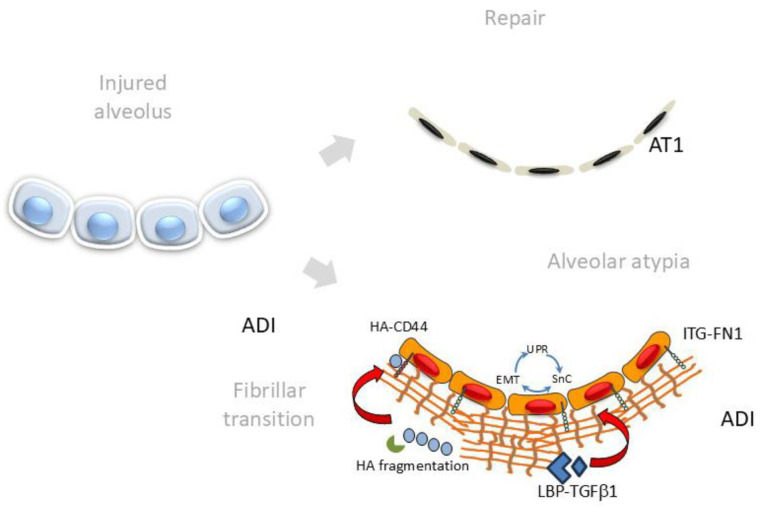
Bidirectional matricellular signaling in alveolar remodeling. Schematic view of alternative pathways for alveolar injury–repair. After ADI recruitment, fibrillar ECM transition provides an environment for sustained activation of the integrated cell stress response.

**Figure 6 ijms-26-05393-f006:**
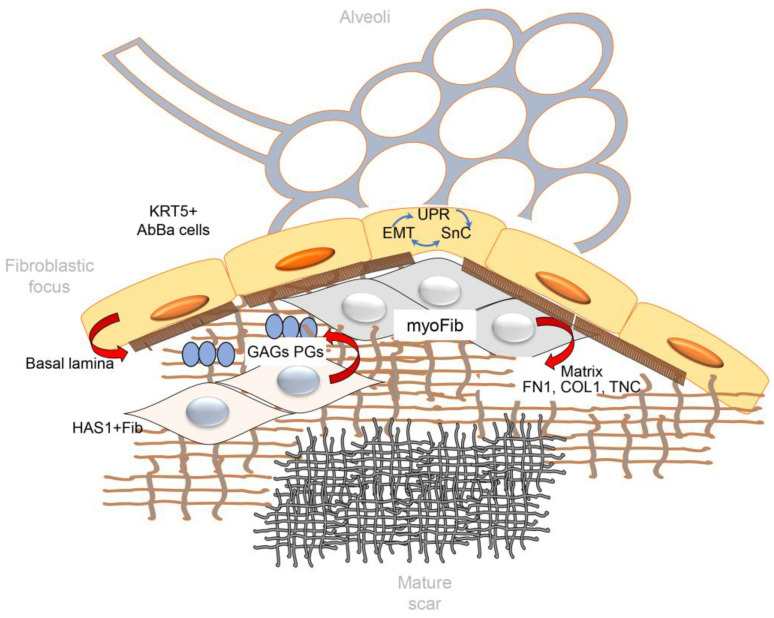
Matricellular interactions driving progression of FF. Schematic view of FF invading an alveolar airspace. Overlying the fibroblast core is a layer of aberrant basaloid cells with activation of the integrated cell stress response. Maturation of the FF matrix through post-translational modifications including collagen crosslinking results in an acellular mature scar.

**Figure 7 ijms-26-05393-f007:**
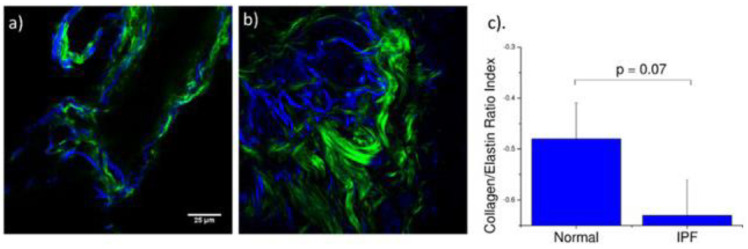
Parenchymal imaging of elastin/collagen index. Single optical sections of (**a**) normal and (**b**) IPF parenchyma, where blue and green is the TPE autofluorescence from elastin and SHG from collagen, respectively. Field of view is 180 µm. (**c**) Averaged collagen/elastin ratio [EV − CV]/[EV + CV] of normal (−0.48) and IPF (−0.63) parenchymal tissues where limiting values of +/− 1 are indicative of all elastin and all collagen, respectively. Normal and IPF parenchyma are statistically different (*p* = 0.07). Adapted from Ref. [154] and used with permission under Creative Commons license.

**Figure 8 ijms-26-05393-f008:**
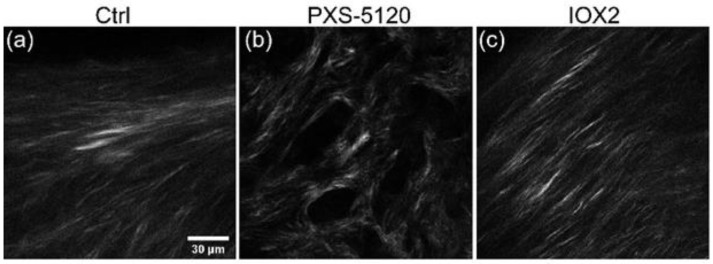
SHG single optical section of IPF spheroids. IPF spheroids were cultured under (**a**) control, (**b**) crosslink inhibition (PXS-5120) and (**c**) crosslink promotion (IOX2). Adapted from Ref. [121] and used with permission under Creative Commons license.

**Table 1 ijms-26-05393-t001:** Gene markers of EMT and senescence.

EMT							
	Mmp9	Fn1	Snai1	Zeb1	Il6	Ldlr	Lif
	Tgfb1	Col1a	Tank	Gadd45a	Gadd45b	Jun	Junb
	Irf1	Ifnl2	Ccl20	Nfkb1			
**Senescence**							
	Angptl4	Areg	Axl	Ccl20	Ccl24	Ccl5	Cdkn1a
	Csf1	Ctnnb	Cxcl1	Dkk1	Edn1	Fgf1	Hmgb1
	Icam1	Igfbp1	Il10	Itga2	Jun	Mmp3	Mmp9
	Nrg1	Plau	Plaur	Rps6ka5	Sema3f	Serpineb3a	Serpine2
	Spp1	Tnf	Vegfa	Vegfc	Vgf	Wnt2	

## Data Availability

No new data were created or analyzed in this study. Data sharing is not applicable to this article.

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
