# Peer review of "Mechanisms on How Matricellular Microenvironments Sustain Idiopathic Pulmonary Fibrosis"

_ijms, 2025, doi:10.3390/ijms26115393_

Round 1
Reviewer 1 Report
Comments and Suggestions for Authors
The review entitled: " Mechanisms How Matricellular Micro-environments Sustain Idiopathic Pulmonary Fibrosis “.The review provides a well-organized Mechanisms How Matricellular Micro-environments Sustain Idiopathic Pulmonary Fibrosis and the study addresses an important gap in the IPE. The study's findings are valuable; however, in each section, several issues need to be significantly addressed.
Abstract:
Line 20: ECM changes. Please add the full name for this abbreviation.
Line 32: PET MRI. Please add the full name for this abbreviation.
Introduction:
- Line 59: Add year of cases (43-63 cases/100,000 people in Westernized countries)
- Mention examples of non-idiopathic forms of fibrosis and what the difference is from IPF
- Discuss other extrinsic and intrinsic risk factors of IPF.
- Discuss briefly the role of genetic predisposition to fibrosis
- It is better to add a paragraph at the end of the background to conclude how the review will explore the current publications on IPE and sections of the current review.
- The Section "Epithelial Microinjury in the Initiation of IPF" is confusing. I expected to read about the role of epithelial microinjury in IPE; however, this section focuses on non-idiopathic forms of fibrosis. Either modify the section title to "Epithelial Microinjury in the Initiation of Fibrosis in general" or focus on IPE.
- Modify all genes name in italic formats.
- Figure 1. Please mention the source of these histological slides. Is it the original work of the researcher, or is it taken from previous research? Are they animal or human slides?
- It is better to add a normal control histology of the lung in Figure 1.
- Discusses if there are other histological pictures or differences with other stains of IPE rather than H&E.
- In section The Unfolded Protein Response (UPR)- (please remove underscore.
- Consider adding a table summarizing different gene expressions that help to illustrate the role of cell-matrix interactions driving alveolar atypia at the end of section 4.
- Page 9, lines 318-322: need in-text citation.
- Page 17 lines 694-700: it is mentioned that our group? Which group and this study were in 2002 (not a recent study -reference 85). Please clarify and modify.
- Add a paragraph on the practical implications and recommendations of your review
Author Response
We thank the individual reviewers for the time and suggestions to improve the quality of our manuscript. We have carefully considered the comments of each and responded appropriately. Specially, we have modified Fig. f1 to include histology of normal human airway, as well as providing clearer outline of the manuscript, a more detailed explanation of epithelial microinjury, and added a section on “Practical implications”.
Reviewer 1:
Comment: In the Abstract, please add the full name for EMT and PET MRI abbreviations.
Response: We have elaborated these common abbreviations.
Comment: In the Introduction, add year of cases (43-63 cases/100,000 people in Westernized countries)
Response: We added the year of this observation (2005) on line 52, Introduction.
Comment: Mention examples of non-idiopathic forms of fibrosis and what the difference is from IPF.
Response: In Section 3, line 184, we state: “While UIP is a prototypical pattern of fibrosis in IPF, UIP itself can be seen as a pattern in other non-idiopathic forms of fibrosis, including connective tissue disease (rheumatoid arthritis and others) or chronic hypersensitivity pneumonitis [31,35,44]. In the setting of hypersensitivity pneumonitis, fibrosis is typically accompanied by a cellular infiltrate and granulomas. However, there is almost always evidence of prominent fibroblastic proliferation [31,32,35].”
Comment: Discuss other extrinsic and intrinsic risk factors of IPF.
Response: We have inserted the following into Section 2: “Factors responsible for epithelial microinjuries that have been identified include latent Epstein-Barr virus, herpesvirus infections, smoking, silicate dust and burn pit exposure (21, 22) Of importance to military personnel, burn pit emissions contain particulate matter combined with toxic gasses and heavy metals (23).”
Comment: Discuss briefly the role of genetic predisposition to fibrosis.
Response: In Section 2, we state: “Genetic predispositions include polymorphisms in cellular senescence pathways, telomere function DNA repair, and wound healing (24-27). In addition, poly-morphisms in mucin processing, have been identified as predisposition to fibrosis (28, 29) as well as missense or deletion mutations in the surfactant protein C (SFTPC) isoforms predisopose to fibrosis. Recent att4ention in mutations in the self-folding/chaperone “BRICHOS” domain of SFPTC have been linked to fa-milial childhood interstitial lung disease (30).”
Comment: It is better to add a paragraph at the end of the background to conclude how the review will explore the current publications on IPF and sections of the current review.
Response: We have added the following under Section 1c. “Review outline”: “
This review will concentrate on the pathological cross-talk between cell-state transitioned epithelial cells and fibroblasts with extracellular matrix. We seek to examine the characteristic features of IPF beginning with the larger context of lung injury, introducing the concept of epithelial microinjury, reviewing the pathological changes of acute lung injury and how these overlap with those characteristic of IPF. We will then examine work that has identifies two distinct matricellular environments important in the pathology of IPF - the alveolar epi-thelial surface and the fibroblastic focus. We will concentrate our analysis on the self-sustaining, autoregulatory cell-matrix interactions occurring within each. We will conclude with examining newer imaging approaches to identify cell state activation and structural changes in ECM and the potential transformative im-pact of these technologies in monitoring IPF progression.
Comment: The Section "Epithelial Microinjury in the Initiation of IPF" is confusing. I expected to read about the role of epithelial microinjury in IPF. however, this section focuses on non-idiopathic forms of fibrosis. Either modify the section title to "Epithelial Microinjury in the Initiation of Fibrosis in general" or focus on IPF.
Response: We have re-written this section to focus on epithelial microinjuries, including the section noted above.
Comment: Modify all genes name in italic formats.
Response: Thank you for this presentation reminder.
Comment: In Figure 1. Please mention the source of these histological slides. Is it the original work of the researcher, or is it taken from previous research? Are they animal or human slides?
Response: these are human samples acquired during routine clinical practice. This is clarified in the Figure 1 legend.
Comment: It is better to add a normal control histology of the lung in Figure 1.
Response: We have added a normal human lung histological image.
Comment: Discuss if there are other histological pictures or differences with other stains of IPF rather than H&E.
Response:
Comment: In section The Unfolded Protein Response (UPR)- (please remove underscore.
Response: Done.
Comment: Consider adding a table summarizing different gene expressions that help to illustrate the role of cell-matrix interactions driving alveolar atypia at the end of section 4.
Response: We have added Table I with a list of the characteristic gene signatures for EMT and cellular senescence.
Comment: Page 9, lines 318-322: need in-text citation.
Response: Done.
Comment: Page 17 lines 694-700: it is mentioned that our group? Which group and this study were in 2002 (not a recent study -reference 85). Please clarify and modify.
Response: Text modified and recent reference updated.
Comment: Add a paragraph on the practical implications and recommendations of your review.
Response: We have added this paragraph (Section 10 in the revised manuscript).
Reviewer 2 Report
Comments and Suggestions for Authors
This is an interesting and comprehensive review about pathophysiology of idiopathic pulmonary fibrosis and the interaction between extracellular matrix and the cells involved in the process
Minor changes.
- Recommend to write in introduction how the manuscript is organized as it starts as 1. Introduction, and then 1a “Background” and 1b”Motivation”, 2. Epithelial microinjury in the initiation of IPF, etc.
- In 1b. Motibation, line 78, the name of the antifibrotic is Nintedanib, please correct.
- Reference 12 cites a clinical trial about use of antifibrotic in Progressive Pulmonary Fibrosis, not exclusively about IPF. You can use reference of the original studies about Nintedanib and Pirfenidone:
- Richeldi L, du Bois RM, Raghu G, Azuma A, Brown KK, Costabel U, Cottin V, Flaherty KR, Hansell DM, Inoue Y, Kim DS, Kolb M, Nicholson AG, Noble PW, Selman M, Taniguchi H, Brun M, Le Maulf F, Girard M, Stowasser S, Schlenker-Herceg R, Disse B, Collard HR; INPULSIS Trial Investigators. Efficacy and safety of nintedanib in idiopathic pulmonary fibrosis. N Engl J Med. 2014 May 29;370(22):2071-82. doi: 10.1056/NEJMoa1402584. Epub 2014 May 18. Erratum in: N Engl J Med. 2015 Aug 20;373(8):782. doi: 10.1056/NEJMx150012. PMID: 24836310.
- King TE Jr, Bradford WZ, Castro-Bernardini S, Fagan EA, Glaspole I, Glassberg MK, Gorina E, Hopkins PM, Kardatzke D, Lancaster L, Lederer DJ, Nathan SD, Pereira CA, Sahn SA, Sussman R, Swigris JJ, Noble PW; ASCEND Study Group. A phase 3 trial of pirfenidone in patients with idiopathic pulmonary fibrosis. N Engl J Med. 2014 May 29;370(22):2083-92. doi: 10.1056/NEJMoa1402582. Epub 2014 May 18. Erratum in: N Engl J Med. 2014 Sep 18;371(12):1172. PMID: 24836312.
- Recommend using the term “fibroblastic foci” instead of “fibroblastic focus”, as it is currently reported in the literature about pulmonary fibrosis and used by pathologists.
Author Response
We thank the individual reviewers for the time and suggestions to improve the quality of our manuscript. We have carefully considered the comments of each and responded appropriately. Specially, we have modified Fig. f1 to include histology of normal human airway, as well as providing clearer outline of the manuscript, a more detailed explanation of epithelial microinjury, and added a section on “Practical implications”.
Reviewer 2:
Comment: This is an interesting and comprehensive review about pathophysiology of idiopathic pulmonary fibrosis and the interaction between extracellular matrix and the cells involved in the process.
Response: Thank you for this assessment.
Minor changes.
Comment: Recommend to write in introduction how the manuscript is organized as it starts as 1. Introduction, and then 1a “Background” and 1b”Motivation”, 2. Epithelial microinjury in the initiation of IPF, etc.
Response: We have added an outline of the Review in Section 1c as noted above.
Comment: In Section 1b, correct the antifibrotic name (Nintedanib).
Response: This typographical error has been corrected..
Comment: use the reference of the original studies about Nintedanib and Pirfenidone:
Response: these citations have been inserted into the text and added to the citation list.
Comment: Recommend using the term “fibroblastic foci” instead of “fibroblastic focus”, Response: the change has been incorporated throughout.
Round 2
Reviewer 1 Report
Comments and Suggestions for Authors
Many thanks for authors all inquiries were addressed.